

# Insights in biomarkers complexity and routine clinical practice for the diagnosis of thyroid nodules and cancer

Maria de Lurdes Godinho de Matos[1], Mafalda Pinto[2], Ana Gonçalves[3], Sule Canberk[2], Maria João Martins Bugalho[4] and Paula Soares[2,5]

[1] Department of Endocrinology, Diabetes and Metabolism, Hospital Curry Cabral, Unidade Local de Saúde São José, Centro Clínico Académico de Lisboa, Lisbon, Portugal
[2] Instituto de Patologia e Imunologia Molecular da Universidade do Porto (IPATIMUP), i3S—Institute for Research & Innovation in Health, Porto, Portugal
[3] Department of Pathology, Unidade Local de Saúde São João, Porto, Portugal
[4] Department of Endocrinology, Hospital de Santa Maria, Unidade Local de Saúde Santa Maria; Medical Faculty, University of Lisbon, Lisbon, Portugal
[5] Medical Faculty, University of Porto, Porto, Portugal

Corresponding authors
Maria de Lurdes Godinho de Matos, lurdesgmatos@gmail.com
Paula Soares, psoares@ipatimup.pt

## ABSTRACT

**Background:** The differential diagnosis between benign and malignant thyroid nodules continues to be a major challenge in clinical practice. The rising incidence of thyroid neoplasm and the low incidence of aggressive thyroid carcinoma, urges the exploration of strategies to improve the diagnostic accuracy in a pre-surgical phase, particularly for indeterminate nodules, and to prevent unnecessary surgeries. Only in 2022, the 5th WHO Classification of Endocrine and Neuroendocrine Tumors, and in 2023, the 3rd Bethesda System for Reporting Thyroid Cytopathology and the European Thyroid Association included biomarkers in their guidelines. In this review, we discuss the integration of biomarkers within the routine clinical practice for diagnosis of thyroid nodules and cancer.

**Methodology:** The literature search for this review was performed through Pub Med, Science Direct, and Google Scholar. We selected 156 publications with significant contributions to this topic, with the majority (86, or 55.1%) published between January 2019 and March 2024, including some publications from our group during those periods. The inclusion criteria were based on articles published in recognized scientific journals with high contributions to the proposed topic. We excluded articles not emphasizing molecular biomarkers in refine the pre-surgical diagnosis of thyroid nodules.

**Results:** We explored genetic biomarkers, considering the division of thyroid neoplasm into *BRAF*-like tumor and *RAS*-like tumor. The specificity of *BRAF* mutation in the diagnosis of papillary thyroid carcinoma (PTC) is nearly 100% but its sensitivity is below 35%. *RAS* mutations are found in a broad spectrum of thyroid neoplasm, from benign to malignant follicular-patterned tumors, but do not increase the ability to distinguish benign from malignant lesions. The overexpression of miRNAs is correlated with tumor aggressiveness, high tumor node metastasis (TMN) stage, and recurrence, representing a real signature of thyroid cancer, particularly PTC. In addition, associations between the expression levels of selected miRNAs and the presence of specific genetic mutations have been related with aggressiveness and worse prognosis.

**Conclusions:** The knowledge of genetic and molecular biomarkers has achieved a high level of complexity, and the difficulties related to its applicability determine that their implementation in clinical practice is not yet a reality. More studies with larger series are needed to optimize their use in routine practice. Additionally, the improvement of new techniques, such as liquid biopsy and/or artificial intelligence, may be the future for a better understanding of molecular biomarkers in thyroid nodular disease.

## INTRODUCTION

Thyroid nodular disease continues to be a major challenge in clinical practice because the differential diagnosis between benign and malignant nodules remains fraught with many difficulties and doubts. Thyroid nodules are commonly found in the general population, with prevalence rates varying based on the method of detection: 2–6% by palpation, 19–35% by ultrasound, and 8–65% based on autopsy data (*Tamhane & Gharib, 2016*). Thyroid cancer (TC) represent about 1% of total cancers, being the most frequent endocrine malignancy with a frequency of more than 90% (*Siegel et al., 2021*; *Miranda-Filho et al., 2021*). Thyroid cancers are diagnosed in 1% to 5% of thyroid nodules (*Grussendorf, Ruschenburg & Brabant, 2022*), being more than 90% differentiated thyroid carcinoma (DTC), out of which papillary thyroid carcinoma (PTC) is the most prevalent (over 85%) (*Li, Brito & Vaccarella, 2020*). The 2022 5th edition of the WHO Classification of Endocrine and Neuroendocrine Tumors (*WHO, 2022*) classified thyroid tumors into two groups based on their cells of origin: the major group derived from follicular epithelial cells and another from calcitonin-secreting C cells (3–5%). Follicular cell-derived tumors, which represent the majority of thyroid neoplasms, are categorized into benign, low-risk, and malignant neoplasms. These tumors show a significant correlation between histological phenotype and genotype, allowing for a refinement of the 5th WHO Classification of Thyroid Tumors. This refinement aims to enhance the understanding and standardization of differentiated thyroid carcinomas. The classification is clinically relevant as it also serves as a prognostic risk classification (Table 1). The incidence of PTC is increasing, attributed to environmental factors and the higher sensitivity of complementary diagnostic techniques, although this increase is not accompanied by a change in mortality rate (*Ferlay et al., 2019*; *Kitahara & Sosa, 2020*). PTC generally has a favorable prognosis, but certain clinicopathological features and genetic alterations can affect its progression (*Ren et al., 2018*). Overdiagnosis of thyroid cancer is currently a global public health issue (*Pizzato et al., 2022*). Improving pre-surgical diagnostic tools is crucial for optimizing treatment and surgical options in thyroid nodular disease (*Hlozek et al., 2022*).

**Table 1  The 2022 WHO classification of follicular cell derived thyroid neoplasm.**

| | Outcome |
|---|---|
| 1. Benign tumors | |
| Thyroid follicular nodular disease | |
| Follicular adenoma | |
| Follicular adenoma with papillary architecture | |
| Oncocytic adenoma of the thyroid | |
| 2. Low-risk neoplasms | |
| Noninvasive follicular thyroid neoplasm with papillary-like nuclear features | |
| Thyroid tumors of uncertain malignant potential | |
| Hyalinizing trabecular tumor | |
| 3. Malignant neoplasms | Outcome |
| Follicular thyroid carcinoma | Favorable |
| Invasive encapsulated follicular variant papillary carcinoma | |
| Papillary thyroid carcinoma | |
| Oncocytic carcinoma of the thyroid | |
| Follicular-derived carcinomas, high-grade<br>   Differentiated high-grade thyroid carcinoma<br>(necrosis and mitosis)<br>   Poorly differentiated thyroid carcinoma<br>(Turin criteria) | Intermediate |
| Anaplastic follicular cell–derived thyroid carcinoma | Poor |

**Note:**
Adapted from: (WHO, 2022; Tamhane & Gharib, 2016).

Ultrasound-guided fine needle aspiration cytology (US-FNAC) remains the gold standard for pre surgical diagnosis of thyroid nodular disease, though over 25% of nodules still do not have a definitive diagnosis (Ali & Cibas, 2018). The 2023 Bethesda System for Reporting Thyroid Cytopathology (TBSRTC) has updated the standardized reporting system for thyroid fine needle aspirations. However, diagnostic limitations persist in nondiagnostic (B-I) and indeterminate Bethesda categories (B-III and B-IV) (Ali et al., 2023). Each category presents a different risk of malignancy (ROM), with varying treatment and follow-up options (Bongiovanni et al., 2018; Basolo et al., 2023).

In recent years, significant advancements have been made in understanding the molecular mechanisms and signaling pathways of thyroid carcinogenesis. Thyroid neoplasms develop and progress based on the activation or inactivation of somatic mutations and changes in gene expression patterns. Key genetic alterations in thyroid cancer include point mutations in the *BRAF*, *RAS*, *TERTp*, *RET*, and *TP53* genes, as well as fusion genes like *RET/PTC*, *PAX8/PPAR-γ*, and *NTRK*.

Numerous studies have aimed to identify genetic biomarkers to improve pre-surgical diagnosis, especially in indeterminate nodules, and to predict thyroid cancer prognosis, but definitive evidence is still lacking (Grimmichova et al., 2022; Nixon et al., 2017). The assessment of microRNA (miRNA) expression is another promising area in thyroid

carcinogenesis research (*Silaghi et al., 2021*; *Titov et al., 2020*; *Agarwal, Bychkov & Jung, 2021*). miRNAs regulate about 30% of the human genome and play a crucial role as gene global regulators of expression (*Wójcicka, Kolanowska & Jażdżewski, 2016*). They influence intracellular regulatory processes involved in carcinogenesis, such as differentiation, proliferation, and apoptosis (*Ludvíková, Kalfeřt & Kholová, 2015*; *Abdullah et al., 2019*). Studies on miRNA expression profiles in human tumors have identified specific "signatures" associated with diagnosis, staging, prognosis, and treatment response (*Jin et al., 2019*; *Marini, Luzi & Brandi, 2011*). Additionally, correlations between the expression levels of selected miRNAs and specific genetic mutations have been linked to increased aggressiveness and poorer prognosis in thyroid cancer. Often, thyroid surgery is the only diagnostic approach for suspicious and malignant nodules. The inability to differentiate between benign and malignant nodules in a pre-surgical phase does not prevent unnecessary surgeries and potential complications (*Hauch et al., 2014*).

Artificial intelligence (AI) has emerged recently as a new area of interest in scientific research, being a complementary tool in the diagnosis and classification of thyroid nodules, mainly in image, in cytology and histology studies, but still with uncertain practical applicability (*Ludwig et al., 2023*; *Sorrenti et al., 2022*).

In the last 2 years, three articles were published with significant guidelines to the approach to thyroid nodular pathology and cancer, which included genetic and molecular biomarkers for pre-surgical diagnosis and prognosis: in 2022, the 5th edition of the WHO Classification of Endocrine and Neuroendocrine Tumors (*WHO, 2022*), and in 2023, the 3rd TBSRTC (*Ali et al., 2023*) and the European Thyroid Association guidelines for the management of thyroid nodules (*Durante et al., 2023*). However, in addition to the lack of consensus on which panels of genetic and molecular biomarkers to use, their implementation in clinical practice is not yet possible in the vast majority of countries due to technical and economic reason (*Oczko-Wojciechowska et al., 2020*; *Moore et al., 2021*). The rising incidence of thyroid neoplasm (*Sung et al., 2021*; *Raposo et al., 2017*; *Vaccarella et al., 2016*), coupled with the low incidence of aggressive thyroid carcinoma (*Durante et al., 2018*), urges the exploration of strategies to improve the diagnostic accuracy in a pre-surgical phase, particularly for indeterminate nodules, to prevent unnecessary surgeries. The aim of this review is to discuss the impact of biomarkers in the routine diagnostic practice of thyroid nodules and cancers, originated from follicular epithelial cells.

## METHODOLOGY

The literature search for this review was performed through the digital platforms Pub Med (https://pubmed.ncbi.nlm.nih.gov/), Science Direct (https://www.sciencedirect.com/), Google Scholar (https://scholar.google.com/). Boolean operators (AND, OR) and predetermined keywords related with molecular biomarkers (genetic and miRNAs), thyroid nodules and cancer, ultrasound fine-needle aspiration cytology (US-FNAC), and Bethesda and WHO classification were applied. We selected 156 publications with significant contribution to this topic, being the majority 86 (55.1%) between January 2019 to March 2024, due to the greater volume of publications in the last 5 years resulting from

increased scientific interest in the subject, 45 (28.8%) between January 2014 to December 2018 and 24 (15.4%) before 2014, including some publications of our group in those periods. The inclusion criteria used for the selection of article titles in our review were based in articles on the same specific topic, published in recognized scientific journals in digital platforms with high contribution to developing the proposed topic. We only included articles that had full text available and described the exact method and results regarding molecular biomarkers and thyroid nodules and cancer. We excluded articles that were not focus straightly on our topic, especially those articles that not emphases the impact of molecular biomarkers in refine the pre-surgical diagnosis of thyroid nodules.

## THYROID NODULES—A GLOBAL PUBLIC HEALTH PROBLEM

Thyroid nodules (TNs) are becoming more prevalent in clinical practice in recent years, with prevalence in the general population varying from 2% to 65%, depending on diagnostic techniques (*Tamhane & Gharib, 2016*). *Mu et al. (2022)* presented a systematic review and meta-analysis about epidemiology of TNs, considering a prevalence of 24.83% (95% CI [21.44–28.55]). They showed that TNs had become more prevalent during 2012–2022 (29.29%) when compared with 2000–2011 (21.53%, $p = 0.02$). Despite the lack of evidence with iodine nutrition, their results showed a significant association between increasing TN prevalence and older age, gender (female preponderance) and overweight. The incidence of TC has increased in many countries all over the world, but mortality continuous stable at lower rates, probably associated to the over diagnosis of TNs (*Pizzato et al., 2022*; *World Health Organization, 2024*). The GLOBOCAN 2022 (*World Health Organization, 2024*) database showed the global distribution of TC incidence and mortality rates in world population. The incidence rates, using the age-standardized rate, were 13.6 per 100,000 women and 4.6 per 100,000 men, and mortality rates were 0.53 for women and 0.35 per 100,000, under one per 100,000 in most countries. The incidence rates in women differ by 15 times among regions of the world, whereas mortality rates were relatively similar across different settings. Incidence rates were five times higher in countries with high human development index, which may be related to the health care system and TC overdiagnosis in each country, and also to other unclear reasons (*Lamartina et al., 2022*; *Li, Dal Maso & Vaccarella, 2020*). TC was the fifth most frequent cancer in women and the thirteen in men in 2022. The numbers of TNs and TC are expected to rise during the next decades, and research is needed to quantify the determinants of these trends, as well as the potentially deleterious effects of overdiagnosis and overtreatment in the world population.

## FINE NEEDLE ASPIRATION CYTOLOGY OF THYROID NEOPLASM

The differential diagnosis of thyroid nodules integrates clinical evaluation, with a history and physical examination, laboratory tests, neck ultrasound (US), and cytology *via* ultrasound-fine needle aspiration (US-FNAC). Thyroid US evaluate some features suggestive of malignancy in nodules, such as hypoechogenicity, irregular/infiltrative border, microcalcifications, central vascularization, and a taller-than-wide shape in the

transverse view, especially if cervical lymph node involvement is present. The diagnostic sensitivity and specificity of each features is variable and none of them alone, can reliably distinguish malignant from benign nodules (*Kobaly, Kim & Mandel, 2022*). Thyroid imaging reporting and data systems (TIRADS) available were created based in risk stratification systems, to standardize the criteria used in thyroid US. However, no consensus exists yet, and current efforts intent to develop an international TIRADS to unify criteria and recommendations (*Dobruch-Sobczak et al., 2022*). US-FNAC, a minimally invasive technique with low complications and reduced costs, is the gold standard method to evaluate thyroid nodules. *Martin & Ellis (1930)*, in 1930, were the first to describe the use of needle biopsy for the diagnosis of thyroid nodules, using an 18-gauge–needle aspiration technique. Thereafter, other techniques using cut-needles were used, but not wide accepted due to the fear of malignant implants in the needle track, false-negative results, and severe complications. In the 1960s, Scandinavian investigators introduced small needle aspiration biopsy of the thyroid, which came into widespread use after the 1980s (*Bäckdahl et al., 1987*). The accuracy of FNAC was improved with the integration of thyroid US in the 2000s. Thyroid US is recommended in patients with a single or a multinodular goiter or in a patient suspected of having a nodule, but it is not to be used as a screening test for the general population (*Gharib et al., 2010*). The US-FNAC technique used is minimally invasive, being a reliable and usefulness diagnostic test for determining malignancy in a nodule, becoming a standard test, often performed by an endocrinologist (*Godinho-Matos, Kocjan & Kurtz, 1992*; *Todsen et al., 2021*; *Jasim, Dean & Gharib, 2023*). FNAC is also required if a suspicious lymph node (LN) was detected by US, and a washout for thyroglobulin measurement should be performed (*Patel et al., 2020*). The need to clarify the terminology concerning thyroid fine needle aspirations, and the communication among physicians, and other health care providers, determined the standardization of criteria. The Bethesda system for reporting thyroid cytopathology (TBSRTC) emerged in 2007, the 1st edition, and in 2016, the 2nd edition and in 2023, the 3rd edition (*Ali & VanderLaan, 2023*). The 3rd TBSRTC, integrating the 5th WHO classification of thyroid tumors (*Ali et al., 2023*), standardized and reporting system for thyroid fine needle aspirations, considered six diagnostic categories, with a different risk of malignancy each, as follow: (I) Nondiagnostic (ND); (II) Benign; (III) Atypia of undetermined significance (AUS); (IV) Follicular neoplasm (FN); (V) Suspicious for malignancy; and (VI) Malignant (Table 2). Bethesda II and VI cytological results have excellent performance when assessing the benign or malignant nature of a thyroid nodule, with a sensitivity and specificity, near to 100% in the diagnosis of thyroid cancer (*Haugen et al., 2017*). Diagnostic limitations are particularly significant in the nondiagnostic (B-I) and in indeterminate Bethesda categories (B-III and B-IV) (*Basolo et al., 2023*). The differential diagnose between benign or malignant thyroid nodules is obtained in 70–75% of cases by US-FNAC (*Nikiforova et al., 2018*), with an accuracy of 68.8%, including 25% of indeterminate nodules, and a positive predictive value (PPV) of 98.6% and a negative predictive value (NPV) of 55.9%, for adult population (*WHO, 2022*). The expected ROM for Bethesda categories was of 13–30% (B-III), 23–34% (B-IV), 67–83% (B-V) and 97–100% (B-VI) (*Lebrun & Salmon, 2024*). When the noninvasive follicular thyroid

**Table 2  The 2023 TBSRTC for adult population with the risk of malignancy (ROM) for each category.**

| Diagnostic category | ROM, mean % (range) | Usual management |
|---|---|---|
| Nondiagnostic | 13 (5–20) | Repeat FNA with ultrasound guidance |
| Benign | 4 (2–7) | Clinical and sonographic follow-up |
| Atypia of undetermined significance | 22 (13–30) | Repeat FNA, molecular testing, diagnostic lobectomy, or surveillance |
| Follicular neoplasm | 30 (23–34) | Molecular testing, diagnostic lobectomy |
| Suspicious for malignancy | 74 (67–83) | Molecular testing, lobectomy or near-total thyroidectomy |
| Malignant | 97 (97–100) | Lobectomy or near-total thyroidectomy |

Note:
TBSRTC, Bethesda System for Reporting Thyroid Cytopathology. Adapted from *Ali & VanderLaan (2023)*, *Gharib et al. (2010)*.

neoplasm with papillary like nuclear features (NIFTP) is excluded, the ROM evaluated changed to 16% (B-III), 23% (B-IV), 65% (B-V) and 94% (B-VI) (*Gharib et al., 2010*). Both the 2023 Bethesda Classification and the 2022 WHO Classification of Thyroid Tumors, consider the analysis of biomarkers to refine the diagnosis in a pre-surgical phase and to provide prognostic information in thyroid tumors. This contributes to reduce unnecessary surgeries, often associated with hypothyroidism and lifelong thyroid hormone replacement, as well as surgical complications, such as hypoparathyroidism (transient or permanent in 0.12–11%, respectively), and recurrent laryngeal nerve paralysis (*Yazıcıoğlu et al., 2021*; *Brisebois et al., 2021*).

# BIOMARKERS IN THYROID NEOPLASM

The study and identification of genetic and molecular alterations in thyroid may contribute with preoperative biomarkers to precise the ROM, and with postoperative molecular testing to provide prognostic information and potential targeted molecular alterations, particularly useful for therapy of radioiodine resistant cases, and contributing to a personalized and effective treatment of the patients (*Acquaviva et al., 2018*; *Baloch et al., 2022*).

## Genetic biomarkers

Knowledge of molecular mechanisms implicated in thyroid carcinogenesis has been attained in recent years. Thyroid neoplasm result from alterations in gene expression patterns, which occur due to a gradual accumulation of genetic and epigenetic events. These changes are associated with specific tumor phenotypes and are implicated in disease etiology. Molecular alterations induce the activation of different signaling pathways, such as the mitogen-activated protein kinase (MAPK), and phosphatidylinositol 3-kinase (PI3K/AKT/mTOR), which are involved in and promote carcinogenesis (*Hsiao & Nikiforov, 2014*). In a few years, the knowledge of molecular mechanisms implicated in thyroid carcinogenesis changed from understanding signaling pathways and identification of a few genes mutations to the knowledge of the main genes implicated in thyroid carcinogenesis, reviewed by *De Leo et al. (2024)*. Genetic changes in thyroid neoplasms were divided in early/driver molecular alterations and late/progression events. Late/progression events may be associated with early/driver molecular alterations and represent the evolution from well-differentiated to high-grade and undifferentiated carcinoma, being

restricted to the less differentiated portions of the tumor. Both early/driver and late/progression molecular alterations can be identified in molecular analysis of cytology FNA in pre-surgical diagnosis. Mutation burden is an additional indicator of genetic instability and is lower in conventional papillary carcinoma, intermediate in aggressive/ advanced papillary and follicular carcinoma, and highest in anaplastic carcinoma (*Pozdeyev et al., 2018*). Most frequent gene mutations present in follicular-cell derived thyroid tumors are *BRAF*, *RAS*, and *TERTp* mutations, associate with clinically relevant clinicopathologic features, as shown in Table 3.

### Oncogenic signaling pathways in thyroid tumors

Signaling pathways have a fundamental role in the regulation of cell growth, proliferation, differentiation, migration and apoptosis through the regulation of gene expression (*Hsiao & Nikiforov, 2014*). Members of signaling pathways are signal transducers between the extracellular environment and the cell's nucleus (*Cantwell-Dorris, O'Leary & Sheils, 2011*). Molecular alterations of the MAPK pathway in TC affects signaling molecules such as *RAS*, *BRAF* and RET (rearranged during transfection)/PTC, a single-pass transmembrane tyrosine kinase receptor. An external stimulus, such as a growth factor, interacts with its receptor on the cell membrane, activating *RAS* (Ras proto-oncogene, GTPase). This triggers a cascade of cellular alterations, as *RAS* activate RAF (Raf proto-oncogene, serine/ threonine kinase) proteins, followed by phosphorylation and activation of MEK (MAPK kinase), and the activation of extracellular-signal-regulated kinase (ERK), which migrates into the nucleus and induce a range of cellular processes. Mutations in genes coding for MAPK pathway proteins are present in about 30% of all human cancers (*Cantwell-Dorris, O'Leary & Sheils, 2011*). The MAPK pathway is crucial in TC development, particularly in papillary (PTC) and follicular thyroid carcinomas (FTC), the two most frequent follicular epithelial cell subtypes. *RAS*, PIK3CA and AKT1 activating mutations, as well as mutations in PTEN, are crucial for activating the PI3K/AKT pathway, especially in FTC (*Prete et al., 2020*). Based on mutational and transcriptomic profiles, PTC is considered a *BRAF*-like tumor while follicular patterned lesions are *RAS*-like tumors. These tumors are associated with different levels of MAPK signaling pathway activation, with stronger activation in *BRAF*-like tumors than in *RAS*-like tumors, which also result in the activation of the PI3K/ AKT signaling pathway (*Papp & Asa, 2015*; *Yoo et al., 2016*).

### BRAF-like tumors

*BRAF*-like tumors are typically represented by PTC, usually with a papillary morphological pattern, except in the invasive encapsulated follicular variant papillary carcinoma (IEFVPTC), separated from other subtypes of PTCs, which is considered *RAS*-like tumor (*WHO, 2022*). *BRAF*-like molecular alterations include *BRAF* V600E mutation and ALK, RET, NTRK1/3 and MET fusions (*Bellevicine et al., 2020*). *BRAF* encodes a serine–threonine kinase of the MAPK signaling pathway (*Tavares et al., 2016*). When mutated, this gene promotes the activation of the MEK–ERK–MAP kinase pathway and tumor development (*Davies et al., 2002*), being present in a wide range of tumors such as melanoma (40–70%), thyroid (45%) and colorectal cancer (10%)

**Table 3 Summary of clinically relevant clinicopathologic implications *of BRAF, RAS and TERTp* gene mutations in follicular-cell derived thyroid tumors (adapted from *De Leo et al. (2024)*).**

| Genes | Molecular pathology | Thyroid tumor type | Clinical and pathological implications |
|---|---|---|---|
| A. Summary of clinically relevant clinicopathologic implications *of BRAF* gene mutations in follicular-cell derived thyroid tumors (adapted from *De Leo et al. (2024)*). | | | |
| BRAF | *BRAF* is a serine/threonine kinase of the MAPK signaling pathway that regulates cell differentiation, proliferation, and survival<br><br>Activating mutations | Papillary carcinoma (40–80%) : *BRAF* p. V600E is more frequent in conventional papillary carcinoma, i.e., classic papillary carcinoma and other subtypes, except the Encapsulated follicular variant of papillary carcinoma that belongs to the *RAS*-like mutated group | *BRAF* p.V600E is used as a diagnostic marker for preoperative fine needle aspiration in molecular typing of cytologically indeterminate thyroid nodules |
| | Hotspot mutations in exon 15; the most common substitution (95% of cases) is c.1799 T > A (p.V600E). *BRAF* p.V600E is the prototype of the *BRAF* p.V600E-like tumor group (2014 TCGA molecular classification scheme) | High-grade non-anaplastic carcinoma of follicular cells: high-grade papillary carcinoma, often belonging to aggressive papillary carcinoma subtypes (*e.g.,* tall cell, hobnail) (<5%)<br><br>PDTC (<5%) | Prognosis: *BRAF* p.V600E has been linked to extra thyroid extension and to an increased risk of PTC recurrence. However, its overall relevance for risk stratification is probably limited when other parameters like stage and histologic typing are taken into account |
| | Other mutations include c.1801A > G (p. K601E) and small deletions or insertions close to codon 600; these mutations generally are present in the *RAS*-like mutated tumor group (2014 TCGA molecular classification scheme). | Anaplastic carcinoma (35%) | Therapy: Anaplastic and aggressive radioactive iodide refractory *BRAF* p.V600E mutated thyroid carcinomas respond to a combination of RAF and MEK kinase inhibitors |
| | AKAP9: *BRAF* | Papillary carcinomas that are radiation-induced that develop more frequently in children | |
| B. Summary of clinically relevant clinicopathological implications *of RAS* genes mutations in follicular-cell derived thyroid tumors (adapted from *De Leo et al. (2024)*). | | | |
| RAS | *HRAS*, *KRAS*, and *NRAS* are G-proteins that have a critical role in the intracellular transduction of signals from the cell membrane. | Follicular adenoma (20–40%)<br><br>Follicular carcinoma (30–50%) | *RAS* mutations can be present in any follicular patterned thyroid nodule or tumor |
| | In tumors of follicular cells, mutations most frequently affect *NRAS* codon 61. | Encapsulated follicular variant papillary carcinoma, invasive and non-invasive (*i.e.,* NIFTP) (25–45%) | The presence of clonal *RAS* mutation is considered a molecular evidence of neoplasm, but may be present in a small number of histologically hyperplastic nodules of follicular nodular disease/ multinodular hyperplasia/thyroid goiter |
| | *RAS* mutations are the prototype of the *RAS*-like mutated tumor group (2014 TCGA molecular classification scheme) | Poorly differentiated carcinoma (20–50%)<br><br>Anaplastic carcinoma (10–50%) | |
| | | Medullary carcinoma (10–20% of sporadic cases, rare in cases with germline RET mutation) | *RAS* mutations are used in preoperative cytology via FNA, particularly in indeterminate thyroid nodules.<br><br>*RAS* mutations indicate that the nodule is neoplastic, but not necessarily malignant, being most *RAS*-mutated nodules diagnosed as benign after surgery: hyperplastic nodule, follicular adenoma, or NIFT |

(Continued)

| Genes | Molecular pathology | Thyroid tumor type | Clinical and pathological implications |
|---|---|---|---|
| **Table 3** (continued) | | | |
| C. Summary of relevant clinicopathological implications *of TERTp* genes mutations in follicular-cell derived thyroid tumors (adapted from *De Leo et al. (2024)*). | | | |
| TERT | TERT is expressed in germ cells and somatic stem cells. It is not normally expressed (or is expressed at very low levels) in most somatic cells. | Follicular carcinoma (10–35%) | "Late" molecular event associated with tumor progression |
| | | Papillary carcinoma, conventional (5–15%) | *TERT* promoter mutations are clonal and highly prevalent in aggressive carcinomas, while they are uncommon and often subclonal in conventional papillary and follicular carcinoma. |
| | | Encapsulated follicular variant papillary carcinoma (5–15%) | |
| | TERT C228T (hotspot position −124) is much more common than TERT C250T (hotspot position −126). | High-grade non-anaplastic carcinoma of follicular cells (20–50%), equally common in poorly differentiated carcinoma and high-grade papillary carcinoma | |
| | Mutations create a novel binding site for transcription factors of the ETS family, which increase TERT transcription and telomerase expression | Anaplastic carcinoma (30–75%) | Some studies indicate that tumors with concomitant presence of *TERT* promoter and with *RAS* and *BRAF* p.V600E mutation have worse prognosis |
| | | | Powerful indicator of poor prognosis and risk for distant metastasis in carcinomas of follicular cells, independent of histologic typing/subtyping and other relevant clinicopathologic parameters |

(*Cantwell-Dorris, O'Leary & Sheils, 2011*). *BRAF* gene is a member of RAF kinase family (ARAF, *BRAF* and CRAF) (*Xing, 2005*). The most common point mutation in *BRAF* arises from a thymine to adenine transversion in nucleotide 1799, causing a substitution of a valine by a glutamic acid at position 600 in exon 15 (Val600Glu), representing 95% of all *BRAF* mutations found in PTC (*Abdullah et al., 2019*; *Pisapia et al., 2020*). In PTC, *BRAF* mutations are the most common molecular alteration, present in 36–69% of cases (*Zolotov, 2016*; *Penna et al., 2017*), and in anaplastic thyroid carcinoma (ATC) in 30–40%, suggesting that *BRAF* mutated PTCs can progress to less differentiated thyroid cancers (*Hsiao & Nikiforov, 2014*; *Soares & Sobrinho-Simões, 2011*). *Lu et al. (2017)*, described that 69.6% of PTCs presented genetic alterations that activate the MAPK pathway, with *BRAF* (58%), *KRAS* and *HRAS* (2.9%). Other authors, *Fakhruddin et al. (2017)* revealed that the incidence of *BRAF* mutations is approximately 60% and *NRAS* is 11%. RET fusions (present in nearly 10–20% of PTC) are highly specific of classical PTC with some morphological specificities associated with RET-fused PTC (diffuse sclerosing PTC and post radiation solid/trabecular PTC) (*Abdullah et al., 2019*; *Chu et al., 2020*; *Soares et al., 2003*). NTRK1–3 (3–5% of PTC) and ALK (1–3% of PTC) fusions are also found in classical PTC (*Rivera et al., 2010*; *Yakushina, Lerner & Lavrov, 2018*). Other less common mutations can be found in PTC as *PPM1D, CHEK2, MEK1, PPARG, THADA, LTK, MET*

and *FRFR2* genes mutations (*Abdullah et al., 2019*). TP53 mutations can be rarely found in PTC, being detected in more than 75% of invasive and undifferentiated carcinomas. The presence of TP53 mutations in differentiated carcinomas has been described as a possible sign of subsequent dedifferentiation in ATC (*Landa et al., 2016*).

### RAS-like tumors

*RAS*-like TCs are tumors presenting a follicular pattern, classically represented by the IEFVPTC and the FTC (*Baloch et al., 2022*). FTC represents 5–15% of differentiated thyroid carcinomas (DTC). However, a higher incidence of FTC has been described in iodine-deficiency areas (*Baloch et al., 2022*). *RAS*-like molecular alterations encompass *RAS* (*NRAS*, *HRAS*, and *KRAS*), *BRAF K601E*, *DICER1*, *EZH1*, *EIF1AX*, *PTEN* mutations and *PPARG*, *THADA* gene fusions (*Baloch et al., 2022*). These molecular alterations are present in several thyroid neoplasms including benign, malignant and malignant follicular-patterned tumors, being considered as 'uncertain malignant potential' (*WHO, 2022*). The *RAS* protein belongs to a family of GTPases (guanosine triphosphate) that regulates cell growth *via* the MAPK and PI3K-AKT pathways. *RAS* gene family include *NRAS*, *HRAS* and *KRAS*, all described mutated in TC. *HRAS* is located at 11p15.5, *KRAS* at 12p12.1 and *NRAS* at 1p13.2 genomic regions (*Zolotov, 2016*; *Tidyman & Rauen, 2016*). They encode highly related G proteins, placed at the inner surface of the cell membrane and diffuse signals arising from the cell membrane TKR and G-protein–coupled receptors along the MAPK, PI3K/AKT and other signaling pathways (*Abdullah et al., 2019*). Mutated *RAS* are described as the second most common alteration in thyroid cancer (*Zolotov, 2016*). *RAS* activating mutations arise in codons 12, 13, 61. The most frequent mutations in thyroid affect codon 61 of *NRAS* (about 95%) and less commonly codon 61 of *HRAS*. For *KRAS*, about 66% of the mutations affect codon 12 and the remaining affect codon 61 (*Schulten et al., 2013*). Several studies associate *RAS* mutations in TC with a better prognosis (*Hsiao & Nikiforov, 2014*; *Volante et al., 2009*). *RAS* mutations affects benign and malignant lesions and were shown to be more frequent in follicular thyroid adenomas (FTA) (26%), FTC (30–50%), FV-PTC (30–45%), poorly differentiated thyroid carcinoma (PDTC) (33%) and ATC (20–40%) (*Hsiao & Nikiforov, 2014*; *Radkay et al., 2014*). In *FTC*, *RAS* mutations, particularly *NRAS* mutations, followed by *HRAS* and *KRAS* mutations, are found in nearly 50% of cases (*Prete et al., 2020*). PAX8-PPARY is present in 10–40% of FTC, in 5–20% of benign lesions (*e.g.*, FTA), in 0–30% of IEFVPTC, in more than 30% in noninvasive follicular thyroid neoplasm with papillary-like nuclear features (NIFTP), in less than 10% in follicular tumor of uncertain malignant potential (FT-UMP), and rarely in well differentiated tumor of uncertain malignant potential (WDT-UMP) (*Lebrun & Salmon, 2024*; *Acquaviva et al., 2018*). EIF1AX is described in various benign and malignant thyroid neoplasm's such as FTAs (*Lebrun & Salmon, 2024*). The second most common *BRAF* mutation found in TCs is *BRAF K601E* (*Afkhami et al., 2016*), consisting of an adenine to guanine transversion at the position 1801, leading to a lysine to glutamic acid substitution at position 601 (Lys601Glu) (*Torregrossa et al., 2016*; *Insilla et al., 2018*). This mutation has been described in follicular-patterned cancer, particularly within the encapsulated follicular variant of PTC. *BRAF* K601E mutant tumors show better

clinical outcomes, and preoperative *BRAF* K601E identification may provide prognostic information and be useful in clinical management (*Afkhami et al., 2016*).

### hTERT mutations

Telomerase activation has been described as a hallmark of cancer, functioning to extend and preserve telomeric DNA (*Hanahan & Weinberg, 2011*). Telomeres are repetitive non-coding sequences that protect the end of chromosomes from degradation. Telomerase, a ribonucleoprotein with reverse transcriptase activity, promotes the extension of telomeres by adding hexameric 5′-TTAGGG-3′ tandem repeats at the end of the chromosome (*Vinagre et al., 2014*). Telomerase has two subunits: TERT and TERC. TERT (telomerase reverse transcriptase) constitutes the catalytic part of human telomerase. The hTERT (human telomerase reverse transcriptase) gene is located on chromosome 5p15.33, and TERC, the RNA component, provides the template for telomerase elongation (*Pestana et al., 2017*). During each round of cell division of somatic cells, telomeres are shortened, reaching the replication limit, when cell senescence or cell death mechanisms are activated (*Vinagre et al., 2014*). Differentiated normal cells do not express telomerase, but cancer cells reactivate telomerase to achieve immortalization and unrestricted proliferation. Telomerase is present in 80–90% of cancers, and up regulation of TERT is well described (*Pestana et al., 2017*; *Kim et al., 1994*). Several mechanisms for TERT reactivation have been proposed, such as hTERT promoter mutations, epigenetic changes, hTERT amplification (*Pestana et al., 2017*; *Takakura et al., 1999*). High telomerase activity has been reported in thyroid tumors but not in normal thyroid tissues (*Melo et al., 2014*), making hTERT expression a potential biomarker for TC. TERT promoter mutations (*TERTp*) create *de novo* consensus binding sites for the "E26 transformation specific" (ETS) family of transcription factors, which can control TERT transcription, both in the presence or absence of promoter mutations (*Thornton et al., 2022*). *TERTp* mutations were described in several cancer types. *Vinagre et al. (2013)*, reported the presence of *TERTp* mutations in bladder cancer (59%), central nervous system (43%), melanoma (29%) and thyroid cancer (10%), suggesting that these mutations enhance telomerase expression (*Vinagre et al., 2013*; *Pópulo et al., 2014*). These mutations occur in two hotspot positions, −124 and −146 bp upstream of the ATG start site of hTERT. They represent nucleotide changes of G > A (C > T in complementary strand) (*Melo et al., 2014*). *TERTp* mutations have been identified in follicular cell derived TC, such as WDTC, PDTC and ATC. In medullar thyroid carcinoma, benign lesions (FTA, thyroiditis and nodular goiter), or normal thyroid tissue, *TERTp* mutations are rarely detected (*Melo et al., 2014*).

 *Melo et al. (2014)* reported the presence of *TERTp* in 7.5% of PTCs, 17.1% of FTCs, 29% of PDTCs and 33.3% of ATCs, and the importance of these mutations for patient prognosis and consequent management. *Panebianco et al. (2019)* showed similar results with *TERTp* mutations found in 7% of PTC, 18% of FTC and 86% for both PDTC/ATC. Several studies have reported that −124 G > A mutation was more common than the −146 G > A mutation (*Pópulo et al., 2014*; *Panebianco et al., 2019*). The presence of these mutations in TC, particularly in WDTC, appears to be a prognostic indicator with impact

in the management of the patients (*Melo et al., 2014*; *Penna et al., 2018*). *Thornton et al. (2022)* considered the existence of a complex regulatory network of TERT transcription in thyroid tumors and emphasized the need for deeper understanding of the mechanisms of TERT deregulation in thyroid tumors before advancing *TERTp*-specific therapeutic strategies.

## Molecular biomarkers–microRNAs

The assessment of microRNAs (miRNAs) expression represents a promising area of study in cancer. miRNAs are an important class of small non-coding RNAs, approximately 22 nucleotides in length. They play a key role in the posttranscriptional regulation of gene expression by reducing the stability of the target messenger RNA (mRNA) and/or repressing its translation (*Fasoulakis et al., 2020*). miRNAs expression influence intracellular regulation of signaling pathways implicated in carcinogenesis, including differentiation, proliferation and apoptosis (*Fasoulakis et al., 2020*). More than 2000 miRNAs have been identified, and understanding the miRNA expression profile in human tumors has allowed the identification of specific "signatures" associated with diagnosis, staging, prognosis, and response to treatment (*Wójcicka, Kolanowska & Jażdżewski, 2016*; *Ludvíková, Kalfeřt & Kholová, 2015*) in multiple types of cancer, such as thyroid, breast, ovarian, lung, colorectal and chronic lymphocytic leukemia (*Wójcicka, Kolanowska & Jażdżewski, 2016*; *Ludvíková, Kalfeřt & Kholová, 2015*; *Pallante et al., 2006*). miRNAs are stable molecules that can be detected in cytology and histology samples (*Santos et al., 2022*; *Lithwick-Yanai et al., 2017*), and can also be studied in the biological fluids (*Chen et al., 2022*; *Xu et al., 2020*; *Ruiz-Pozo et al., 2023*). miRNAs can act as tumor promoters (oncomiRs) that drive tumorigenesis by targeting tumor suppressor genes or activate oncogenic pathways. Conversely, tumor suppressor miRNAs counteract tumor development by targeting oncogenes or regulating key tumor-suppressive pathways, such as cell cycle arrest and apoptosis (*Fasoulakis et al., 2020*). In TC, alterations in miRNAs biogenesis occur, and oncogenic miRNAs activate the MAPK signaling pathway cascade (*Luzón-Toro et al., 2019*; *Celano et al., 2017*). *Nikiforova et al. (2008)* described the biological and diagnostic value of miRNAs expression in normal and tumor thyroid tissue, noting that miRNAs in thyroid tumor are 32% upregulated and 38% downregulated. miRNA expression is tissue-specific and varies among different types of TC (*Ludvíková, Kalfeřt & Kholová, 2015*). In PTCs, classical miRNAs involved recurrently are miR-146b, miR-221 and miR-222 (*Matos et al., 2024*), upregulated and miR-15a usually down regulated. Upregulated miRNAs have been shown to downregulate proto-oncogene receptor tyrosine kinases (c-KIT), which are involved in cell differentiation and growth, while miR-15a downregulates B-cell lymphoma 2 (BCL-2) and controls apoptosis of PTC through AKT pathway (*Wójcicka, Kolanowska & Jażdżewski, 2016*; *Ludvíková, Kalfeřt & Kholová, 2015*; *Fasoulakis et al., 2020*). *Pallante et al. (2006)* showed that miR-221 and miR-222 are overexpressed in PTC and validated that these miRNAs downregulate c-KIT levels in PTC. *Lee et al. (2013)* identified in tissue samples that the expression of miR-222 was 10.8 times higher, and miR-146b was 8.9 times higher in tumors with recurrence. They analyzed the expression of miRNAs in plasma samples from 42 PTC patients before and

after surgery; miR-146b, miR-221, and miR-222 were significantly upregulated before surgery and downregulated after surgery (*Lee et al., 2013*). *Castagna et al. (2019)* have shown, in a series of fine-needle aspiration cytology samples, that miR-146b, miR-221 and miR-222 are overexpressed in malignant or suspicious of malignancy nodules, confirming the utility of these miRNAs in the classification of thyroid nodules. *Boufraqech et al. (2014)* demonstrated that miR-145 has tissue specific tumor suppressor functions by interacting with AKT3, which regulates the PI3K/Akt pathway. Intriguingly, miR-145 expression is downregulated in PTC tissues, but it is upregulated in the plasma of PTC patients. *In vivo* and *in vitro* studies showed that the upregulation of miR-145 decreases the growth and metastases of thyroid cancer cells (*Boufraqech et al., 2014*). In another study, *Zhou, Tang & Zhou (2019)* analyzing tissue samples from 28 patients with PTC and adjacent non-tumor tissue from the same patients revealed that miR-296-5p expression in PTC was down regulated. However, *Zou et al. (2020)* described that plasma miR-296-5p was overexpressed in PTC and based on the different levels of miR-296-5p expression between plasma and tissue, they considered miR-296-5p a possible biomarker for the diagnosis of thyroid cancer. *Ruiz-Pozo et al. (2023)* reviewed a vast list of expressed miRNAs and genes related to PTC and highlighted the great potential of miRNAs to refine the diagnosis of PTCs in tissue and plasma samples of PTC patients.

## BIOMARKERS IN CLINICAL PRACTICE

Only in 2022, the 5th WHO classification of thyroid neoplasms (*WHO, 2022*) and in 2023 the 3rd TBSRTC (*Ali et al., 2023*) included genetic and molecular biomarkers for pre-surgical diagnosis, refining the diagnostic categories and their risk of malignancy, and for prognosis of thyroid nodules, with corresponding management strategies. Molecular tests are especially recommended for indeterminate cytological FNAs, contributing to management strategies (*Durante et al., 2023*). Currently, many molecular platforms exist for thyroid FNAs molecular analysis, ranging from laboratory-developed tests, like the 7-gene mutational panel used in some European countries (*Bardet et al., 2021*), to commercialized tests such as Afirma, ThyroSeq, ThyGenX/ThyraMIR, and RosettaGX Reveal. These platforms use different targets and present various negative predictive values (NPV) (81–100%) and positive predictive values (PPV) (29–81%) (*Gharib et al., 2010*; *Song et al., 2020*). The SWEETMAC study (2021) (*Bardet et al., 2021*), using the 7-panel mutation testing in indeterminate thyroid nodules, presented an accuracy of 88%, sensitivity of 48%, specificity of 95%, PPV of 67%, and NPV of 91%. The results increased the overall risk of cancer from 16% to 67% in mutated nodules and, by diminishing it to 9% in wild type, confirmed the relevance of the 7-panel mutation testing in the diagnosis of indeterminate nodules. *Tang et al. (2023)* presented a study using a 5-gene panel test on FNAC samples from 759 nodules, reporting sensitivity, specificity, PPV, NPV, and accuracy of 96.83%, 100%, 100%, 42.86%, and 96.90%, respectively. They concluded that the 5-gene panel may represent a valid adjunct technique for distinguishing thyroid nodules. Despite the European and American thyroid associations (*Durante et al., 2023*; *Haugen et al., 2017*) consider the inclusion of molecular tests in US-FNAC, no consensus is yet achieved on the best molecular panel to use in the diagnosis of thyroid nodules.

Available molecular tests have been developed in United States of America, and it is important to assess whether such tests can be used in other populations (*Grimmichova et al., 2022*). The most significant disadvantage of commercially available tests is related to their cost, and the lack of highly specialized reference laboratories in many countries (*Titov et al., 2020*; *Agarwal, Bychkov & Jung, 2021*). Other limitations in the studies reviewed on biomarkers in the diagnosis of thyroid nodular disease were mainly related to their retrospective nature and the small size of the samples, as well as the available laboratory techniques for studying molecular biomarkers in cytology and histology samples. Many doubts persist regarding the reliability of cyto-histological correlation, depending on the technique used and the quality of the cytological and histological samples. Most limitations in cytology samples were caused by the small amount of cellular material, reducing the number of neoplastic cells, and the heterogeneity and multifocality of tumors (*Matos et al., 2024*). The quality of hystology samples stored in pathology archives for many years was affected by the degradation of nucleic acids due to formalin fixation (*Penna et al., 2017*). US-FNAC is the central minimally invasive method for the study of thyroid nodular disease in a pre-surgical phase. An acceptable matched cyto-histological genetic profile agreement for molecular alterations was attained in a study of our group (*Matos et al., 2024*). Using 209 cases of DTC, our results indicate a good level of consistency between the two types of samples in the genetic analysis. *BRAF* and *HRAS* were the genes showing substantial cyto-histological molecular agreement, whereas moderate agreement was found for *NRAS* and *TERTp* genes. *RAS* genes were found to be the most frequently mutated in the indeterminate nodules, in accordance with *Censi et al. (2017)*. The *BRAF* and/or *TERTp* mutations were only present in malignant cases, with a PPV of 100%, in both cytology and histology samples. While the Genetic profile's ability to exclude malignancy was limited (low sensitivity and NPV), it effectively confirmed malignancy in cytology samples, which we could consider as "rule-in" tests, in accordance with 2015 ATA guidelines (PPV > 95%) (*Haugen et al., 2017*), reducing repeated US-FNACs. A good cyto-histological genetic profile agreement for molecular alterations was achieved between the two types of samples in indeterminate nodules, reinforcing the role of molecular analyses before surgery in those cases. The study of miRNA expression can be useful in a precocious and accurate diagnosis, as well as in the evaluation of progression of TC (*Yoruker et al., 2016*; *Galuppini et al., 2022*). The overexpression of miRNAs is correlated with tumor aggressiveness, high tumor node metastasis (TMN) stage, ETE, recurrence, and metastases, representing a real signature of TC, particularly of PTC (*Celano et al., 2017*; *Pamedytyte et al., 2020*), that can be analyzed in cyto-histology samples or in blood (circulating miRNAs) (*Park et al., 2021*; *Pitoia, 2024*).

## THE USEFULNESS OF MOLECULAR BIOMARKERS IN PROGNOSIS

The 5th WHO Classification of thyroid tumors considered new tumor entities that are clinically and morphologically borderline with a very low risk of distant metastases (*Baloch et al., 2022*). Low-risk follicular neoplasms represent a clinical challenge due to a lack of consensus among observers concerning whether tumor morphological characteristics

indicate malignancy, such as the presence of capsular and vascular invasion, and/or extra thyroidal extension (ETE) (*Nishino & Nikiforova, 2018*; *Turk et al., 2019*). Additionally, low-risk neoplasm are not associated with specific diagnostic molecular biomarkers and are considered "molecular indeterminate" tumors. *RAS* mutations are found in a broad spectrum of thyroid neoplasm, from benign to malignant follicular-patterned tumors (*Baloch et al., 2022*). Tumors with *RAS* mutations are associated with a good prognosis (*Tidyman & Rauen, 2016*; *Gilani et al., 2022*). However, there is evidence that *RAS* mutations may be an early event in tumorigenesis, potentially leading to the development of the malignancy and dedifferentiation of the cells (*Radkay et al., 2014*). Detecting *RAS* mutations in thyroid samples in cytology FNAs, particularly in indeterminate nodules, provides improvement on risk stratification with evidence for neoplasia and has the potential for patient management guidance (*Radkay et al., 2014*; *Howell, Hodak & Yip, 2013*). However, it does not help distinguish between benign and malignant lesions. *BRAF* and *RAS* are mutually exclusive events, with no concomitant mutations expected (*Park et al., 2013*). The specificity of *BRAF* mutation in the diagnosis of PTC is nearly 100%, but its sensitivity is below 35% (*Xing, 2005*; *Matos et al., 2024*). Different studies have shown an association between *BRAF*V600E and ETE, tumor multifocality, lack of tumor capsule, and the presence of lymph node metastases, all of which predict a high propensity for local aggressiveness (*Torregrossa et al., 2016*; *Xing et al., 2014*). Although *BRAF* mutations were significantly associated with several clinicopathological features (*Liu et al., 2018*), these mutations are also highly prevalent in low-risk, infracentimeter PTCs, and their prognostic role remains debated (*Soares & Sobrinho-Simões, 2011*; *Scheffel, Dora & Maia, 2022*). Several authors demonstrated that *TERTp* mutations are associated with worse pathological features (older age, tumor size and stage), ETE, vascular invasion, aggressive behavior, treatment resistance, distant metastases, poor prognosis, and a higher risk of disease-specific mortality (*Melo et al., 2014*; *Panebianco et al., 2019*; *Penna et al., 2018*; *Bournaud et al., 2019*). A synergistic impact of *TERTp* mutations with *BRAF* mutations has been suggested, leading to a worse prognosis (*e.g.*, distant metastases, cancer dedifferentiation), particularly in older patients (*Estrada-Flórez et al., 2019*). *TERTp* mutations also impact resistance to radioactive treatment as described in these tumors (*Abdullah et al., 2019*; *Volante et al., 2009*). Moreover, the presence of *TERTp*, PLEKHS1 promoter or TP53 mutations (in association with either *BRAF* or *RAS* mutations) was significantly associated with radioiodine refractory disease (*Jung et al., 2020*). In advanced TC, molecular testing must be considered since molecular alterations are targetable by specific drugs, such as RET, NTRK1-3, *BRAF*, ALK, ROS and mTOR (*Gharib et al., 2010*; *Salvatore, Santoro & Schlumberger, 2021*). Several associations between the expression levels of miR-146b, miR-221 and miR-222 with the presence of *BRAF* mutations in PTCs have been described by many authors, including our group (*Matos et al., 2024*). *Yang & Choi (2020)* showed that miR-146b expression was significantly associated with the presence of *BRAF* mutations. *Lee et al. (2014)*, in a study of 100 PTCs patients, showed that tumors where the *BRAF* V600E mutation was detected concomitantly with miR-146b overexpression had shorter patient survival. In another study, upregulation of miR-146b was associated with tumor aggressiveness and poor clinicopathological features, such as

extrathyroid and capsule invasion, and the presence of lymph nodes or distant metastases (*Sistrunk et al., 2019*). *Sun et al. (2013)* showed that miR-221, miR-222 and miR-146b, together with miR-181, were upregulated in PTCs patients with *BRAF* mutations. These findings suggest a connection or influence of the presence of *BRAF* mutations with miRNAs expression in the development of TC.

## FUTURE PROSPECTS

During the development of this review, we added evidence that molecular markers can be of significant value to daily clinical practice. For that to be realized, however, additional studies and developments are needed to support that practice with robust indicators. The implementation of prospective studies of biomarkers in cytology and histology samples, with larger sample sizes and adapted to different populations, is mandatory to achieve a comprehensive understanding of molecular biomarkers and thyroid tumor features and behavior. The improvement of new techniques, such as liquid biopsy, a non-invasive method that detects and analyzes biological molecules released from the tumor into the bloodstream, is crucial. Different technologies can identify circulating tumor cells, circulating free nucleic acids (cDNA, cfRNA, and miRNA), and tumor-derived extracellular vesicles (EVs) in the serum of cancer patients (*Alix-Panabières & Pantel, 2021*), thus providing valuable molecular information (*van Dessel et al., 2018*). Circulating free-DNA and circulating miRNAs are more stable compared to cfRNA, and easier to investigate (*Pös et al., 2018*; *Vitale et al., 2021*). A set of miRNAs (*e.g.*, miRNA-375, 34a, 145b, 221, 222, 155, Let-7, and 181b) have been evaluated in serum and proposed as diagnostic biomarker to distinguish PTCs from benign nodules and identify cancer at an early stage (*Mahmoudian-Sani et al., 2017*). In a prospective observational study (*Rosignolo et al., 2017*), the serum levels of 754 miRNAs were measured in 11 PTC patients before and after surgery. Among these, miRNA-146 and miRNA-221 were validated as serum tumor biomarkers during post-surgical follow-up, showing a significant correlation with disease recurrence, even in patients with low thyroglobulin levels. In fact, the use of liquid biopsy in thyroid cancer can have a significant role, as it can contribute to accurate patients diagnosis, prognosis, and monitoring tumor recurrence or treatment response (*Romano et al., 2021*). However, the use of liquid biopsy in TC presents several challenges, mainly depending on the sensitivity and specificity of the different methods, and tumor subtypes and stages. Other technical limitations in the use of liquid biopsy relate to the quantity of circulating material retrieved, especially in the context of early-stage disease, and issues of standardization, reproducibility, and validity (*Geeurickx & Hendrix, 2020*). Artificial Intelligence (AI) may be the next step in better understanding molecular biomarkers in thyroid nodular disease by integrating clinical and imaging features with cyto-histological and molecular diagnosis (*Bini et al., 2021*). Interest in the application of AI in thyroidology began in the 1990s, to improve the care of patients with thyroid nodules and cancer, as well as functional and autoimmune thyroid disease (*Toro-Tobon et al., 2023*). *Ludwig et al. (2023)*, suggest its application at various stages in the diagnosis and treatment of thyroid nodules. Application of AI in the diagnostic of thyroid disease includes thyroid ultrasound (*Cao et al., 2023*), cytopathology (*Wang et al., 2022*) or

histopathology studies (*Siller et al., 2022*), and nuclear medicine techniques such as thyroid scintigraphy (*Qiao et al., 2021*; *Currie & Iqbal, 2022*) or SPECT (single-photon emission computed tomography) (*Ma et al., 2019*). AI models may also be used in surgery for preoperative and intraoperative decision-making (*Madani & Feldman, 2019*).

## CONCLUSIONS

The advances in the knowledge of molecular biomarkers for the diagnosis and prognosis of thyroid nodules and cancer reinforce the potential clinical utility of molecular testing in cytological assessments using US-FNA or blood. These advances may anticipate the genetic and molecular profiles of the tumors and their biological behavior. However, the complexity of biomarkers, the lack of consensus on the best panels to use (aiming for better risk stratification), the need for high technology and specialized laboratories, and the high costs not supported by national health care systems in many countries determine that the implementation of biomarkers in clinical practice is not yet a reality. More studies of molecular biomarkers are needed to achieve an international consensus on their applicability for diagnosis, risk stratification and prognosis of thyroid nodules, as well as for their introduction in routine clinical practice. The improvement of new techniques, such as liquid biopsy, will be complementary to tissue biopsy but is still not yet applicable in clinical practice. AI can bring significant benefits for patients, including the reduction of unnecessary FNACs, better surgery options, and reduced costs. However, despite the legal responsibility issues associated with the use of AI, more studies are needed to refine its accuracy in distinguishing between benign and malignant thyroid tumors. By integrating molecular knowledge with clinical evaluation and image features in the work flow of thyroid nodules diagnosis, clinicians may improve the accuracy of their decisions, reducing the number of FNACs, particularly in indeterminate nodules, and may optimize treatment strategies, from surgical options to active surveillance.

### Funding

Sule Canberk received, in the framework of a Ph.D. grant: (SFRH/BD/147650/2019) Portuguese funds through Fundação para a Ciência e a Tecnologia (FCT). This study is part of the project "Institute for Research and Innovation in Health Sciences" (UID/BIM/ 04293/2019) and the project "The Porto Comprehensive Cancer Center" ref. NOR-TE-01- 0145-FEDER-072678—Consórcio PORTO.CCC—Porto. Comprehensive Cancer Center Raquel Seruca. The other authors received no funding for this work. The funders had no role in study design, data collection and analysis, decision to publish, or preparation of the manuscript.

### Grant Disclosures

The following grant information was disclosed by the authors:
Portuguese funds through Fundação para a Ciência e a Tecnologia (FCT): SFRH/BD/

147650/2019.

Institute for Research and Innovation in Health Sciences: UID/BIM/04293/2019.

The Porto Comprehensive Cancer Center: NOR-TE-01-0145-FEDER-072678—Consórcio PORTO.CCC—Porto.

Comprehensive Cancer Center Raquel Seruca.

## Competing Interests

Paula Soares is an Academic Editor for PeerJ. The other authors declare that they have no competing interests.

## Author Contributions

- Maria de Lurdes Godinho de Matos conceived and designed the experiments, performed the experiments, analyzed the data, prepared figures and/or tables, authored or reviewed drafts of the article, and approved the final draft.
- Mafalda Pinto performed the experiments, analyzed the data, prepared figures and/or tables, authored or reviewed drafts of the article, and approved the final draft.
- Ana Gonçalves performed the experiments, analyzed the data, prepared figures and/or tables, authored or reviewed drafts of the article, and approved the final draft.
- Sule Canberk performed the experiments, analyzed the data, authored or reviewed drafts of the article, and approved the final draft.
- Maria João Martins Bugalho analyzed the data, authored or reviewed drafts of the article, and approved the final draft.
- Paula Soares conceived and designed the experiments, analyzed the data, authored or reviewed drafts of the article, and approved the final draft.

## Data Availability

This is a literature review.

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
