# Peer review of "Insights in biomarkers complexity and routine clinical practice for the diagnosis of thyroid nodules and cancer"

_PeerJ, doi:10.7717/peerj.18801_

## Round 0.1 · original submission · Minor Revisions

Dear Professors Matos and Soares,

We have now received 3 reviews of your review paper on biomarkers for thyroid cancer and I am pleased to inform you that all 3 reviewers have recommended just minor revision; thus, we would be happy to publish your paper if you can address all the reviewer comments. You should now have. access to these comments. Reviewer 1 was very general, but seems to have hit on issues that were addressed in more detail by reviewers 2 and 3; therefore, I think you can use the specific points they made as a guide to addressing reviewer 1's comments. Though there will be a bit of reworking needed to accommodate all their comments, they seem reasonable and should make the manuscript more accessible for a wider audience.
Best, Eric Schirmer (re-assigned Academic Editor)

·

Basic reporting

Review is broad but target audiences need to be identified and mentioned.
Authors to correlate if review addresses current trends, or emerging issues of broad relevance across disciplines.

Experimental design

Assessment Tools and study quality can be added.

Validity of the findings

Highlight Unanswered Questions.

·

Basic reporting

General comments
The spelling and punctuation are very good. No issues were detected.
Abstract
The abstract is concise. All the necessary information about the study is included.

Background
- The information provided in the introduction is important for the comprehension of the article.
- The objective of the study is clearly mentioned.

Discussion
- The discussion is of great quality and includes updated data.
- The authors should inform the reader about the study's limitations.
Conclusion
From the presented data, the conclusion is complete and represents the work that the authors did.

Experimental design

Methods
- The methods are sufficiently explained by the authors.

Results
- The results are presented in a very extensive way.
- The tables and figures are really helpful and necessary for the completion of the authors' work.

Validity of the findings

1) "The thyroid gland, consisting of two connected lobes, is one of the largest endocrine glands in the human body, weighing 20 - 30 g in adults. Thyroid lesions are often found on the gland, with a prevalence of 4%–7%. Most of them are asymptomatic, and thyroid hormone secretion is normal."
I would suggest adding this information in the introduction section and consider citing the recently published article:
https://pubmed.ncbi.nlm.nih.gov/32965923/
2) I would like a brief discussion on the Bethesda classification system for reporting thyroid cytopathology ( especially for type II and III) and consider citing the recently published articles on Bethesda II and III:
https://pubmed.ncbi.nlm.nih.gov/33749812/
https://pubmed.ncbi.nlm.nih.gov/34734516/
Which is the percentage of incidental malignancy according to these studies for Bethesda II and III?

Additional comments

3) According to the literature, there is controversy on the selection of the best surgical treatment for differentiated thyroid cancer (TC), total thyroidectomy (TT), and subtotal thyroidectomy (STT). Is there an increased risk of early complications after TT in comparison with STT?
I would like a brief discussion on that and consider citing the related published article:
https://pubmed.ncbi.nlm.nih.gov/36318685/

Reviewer 3 ·

Basic reporting

Overall view: This paper summarizes a vast knowledge on the molecular biomarkers in thyroid cancer diagnosis and nodule management, in reference to the newly published classifications/guidelines of the most relevant institutions in the field (WHO, ETA etc.), which include the molecular alterations in their classifications. I find the paper is comprehensible and relevant to the field.
1. The paper is easy to follow and written in solid English, however, in my opinion, it requires improvement, as there are quite a few grammatical errors (singular vs. plural, prepositions, etc.) and frequent misuse of terms. I recommend editing by a subject matter expert and a native English-speaking editor to correct the grammar, and sentence structure.
2. Figure 1 is not referenced in the text.
3. Reference in Li 313 (79) should be replaced with a more suitable one, dealing with mutations in ATC such as PMID: 26878173. Reference 98 from Li397 and Li398 should be replaced with a citation more related to the general role of microRNAs in gene expression regulation and carcinogenesis.

Experimental design

1. Abstract: Li 35, specify which ETA guidelines included molecular biomarkers. The Results part of the Abstract can be halved or at least shortened, while Li 59 -62 might be more suitable for the Conclusion part.
2. The authors are reporting the number of selected publications in overlapping ranges (Abstract Li 41-44, text Li 143-146). It would be good to have an explanation for this kind of grouping (for example, trying to point out the increase in the volume of publications in the last 5 years, etc). Also, were there specific criteria for the groups with international recognition and high contributions, such as the number of publications per year, h-index, the number of successful international collaborations? If yes, it would be valuable to include this in the Material and Methods, or at least touch upon it and explain in more detail how publications were chosen to be included in the review. At present, the criteria seem a bit too arbitrary.
3. The text from the Li 382-384 should be moved after Li366 and merged with the paragraph that begins at Li367.
4. I would recommend moving the sentence in Li388-390 to the end of the paragraph “Genetic biomarkers”.
5. The section “Biomarkers in clinical practice” should be moved in front of the section “The usefulness of molecular markers in prognosis”, as this is a more logical order, in my view. Also, the “Biomarkers” section can be expanded a bit (several sentences) to include the experiences with a 7-gene mutational panel that was used is some European countries.

Validity of the findings

1. It would be worth including (if available) which molecular tests are recommended for refining the preoperative diagnosis of thyroid tumors, i.e. which are the most common, universally recommended markers to be tested, as this is mentioned in Li 227-229.
2. Given that the topics of liquid biopsy and AI are announced in the Abstract (and the cover letter), in my opinion, the authors should expand more on them to include the current state-of-the-art in thyroid oncology. I find that it is too short in its present form, since, at least for liquid biopsy, there have been many studies regarding the possibility of detecting various species of biomarkers, albeit with slim transfer to the clinic. AI field is very novel, and I believe the readers would benefit from a short overview on the usefulness of this tool in thyroid cancer diagnostics and prognostics. These two matters could be covered in separate paragraphs preceding the Conclusions section.

---

## Round 0.2 · accepted · Accept

Dear Drs de Matos and Soares,

As only minor revisions were requested, I have gone over the reviewer comments and your changes myself. I would note that this would have been easier if you had included a point-by-point rebuttal letter addressing the reviewer's comments and would encourage you to do so in the future as its absence greatly increases the effort required of reviewer/editor approving revisions. I would also note that if you had included this it would have allowed you to include an explanation of why you did not address most of reviewer 2's comments. Since most of the additional references requested were self citations, I decided to waive the reviewer's push for their inclusion and since the review was focused on biomarkers rather than treatment options I decided to waive their request for more of an elaboration on the best surgical treatment for different types of thyroid cancer. Overall, the manuscript reads much better; so the changes you have made have definitely improved it.
Regards, Eric Schirmer